# Sous-Vide as an Alternative Method of Cooking to Improve the Quality of Meat: A Review

**DOI:** 10.3390/foods12163110

**Published:** 2023-08-18

**Authors:** Agnieszka Latoch, Artur Głuchowski, Ewa Czarniecka-Skubina

**Affiliations:** 1Department of Animal Food Technology, University of Life Sciences in Lublin, 8 Skromna St., 20-704 Lublin, Poland; agnieszka.latoch@up.lublin.pl; 2Department of Food Gastronomy and Food Hygiene, Institute of Human Nutrition Sciences, Warsaw University of Life Sciences (WULS), 166 Nowoursynowska St., 02-787 Warsaw, Poland; ewa_czarniecka_skubina@sggw.edu.pl

**Keywords:** sous vide, pork, beef, poultry, cooking yield, sensory quality, nutritional value, microbiological quality

## Abstract

Sous-vide (SV) is a method of cooking previously vacuum-packed raw materials under strictly controlled conditions of time and temperature. Over the past few years, scientific articles have explored the physical, biochemical, and microbiological properties of SV cooking. In this review, we provide a critical appraisal of SV as an alternative method of meat cooking, including the types of methods, types of SV meat products, and effects of SV parameters on the meat quality and the mechanisms of transformation taking place in meat during SV cooking. Based on the available data, it can be concluded that most research on the SV method refers to poultry. The yield of the process depends on the meat type and characteristics, and decreases with increasing temperature, while time duration does not have an impact. Appropriate temperatures in this method make it possible to control the changes in products and affect their sensory quality. Vacuum conditions are given a minor role, but they are important during storage. The limited number of studies on the approximate composition of SV meat products makes it challenging to draw summarizing conclusions on this subject. The SV method allows for a higher microbiological quality of stored meat than conventional methods. The literature suggests that the SV method of preparing beef, pork, and poultry has many advantages.

## 1. Introduction

Sous-vide (SV) is a method of cooking previously vacuum-packed raw materials under strictly controlled conditions of time and temperature. Until 2011, it was mainly used to extend the shelf life of food products. In this case, its usage involves combining methods that integrate several parallel fixing factors: vacuum conditions, low temperature, and extended process time. These factors act additively, thanks to which they limit unfavorable changes more effectively than individually extended process times [1]. Currently, it has become one of the methods of culinary processing used to create new dishes in catering establishments [2]. In the sous-vide method, temperatures usually range from 50 to 85 °C, which requires a longer heating time (from 2 to 48 h) compared to traditional methods (HTST: high temperature, short time). Heat treatment conditions depend on the type of meat, size of the cut meat, intramuscular connective tissue, and myofibrillar protein components [3,4].

Low-temperature sous-vide heat treatment consists of five stages: pre-treatment, including seasoning; preliminary heat treatment to achieve characteristic flavor features (Maillard reactions); vacuum or modified atmosphere packaging; pasteurization under controlled conditions of temperature and time in a water bath or in a convection-steam oven [5,6]; and serving. Sous-vide products can be served immediately after heat treatment, chilled in the packaging to a temperature of 0–3 °C, stored in a refrigerator, and heated before delivery [7]. Chilling and refrigerated storage allow for overproduction in catering enterprises while retaining good nutritional value [8,9] and sensory quality [9,10].

Llave et al. [11] distinguished two varieties of the LT sous-vide method: traditional (LTLT: low temperature, long time) and innovative (LTST: low temperature, short time). The first is to achieve the same doneness inside the entire product. In the second method, due to the reduced process time, meat products with varying degrees of doneness can be obtained. Such products have a better taste, texture (less muscle contraction), and juiciness than those made using a higher process temperature or traditional methods. The latest variation of this method is sequential heat treatment at increasing temperatures (m-SV: multi-stage sous-vide cooking), as well as the sous-vide pressure technique (HPP-SV) or ultrasound-assisted sous-vide (USV) [3,12,13,14,15,16].

The sous-vide product range can be divided into four main categories: 90/10 heat treated, 70/2 heat treated, cooked to a minimum core temperature of 63 °C, and lightly processed. Sous-vide products are subjected to 90/10 heat treatment until the thickest part reaches 90 °C for 10 min (or its equivalent), aiming to reduce 99.9999% of the non-proteolytic *Clostridium botulinum* strains. This treatment is recommended for retail and extended shelf-life products. The 70/2 heat treatment reduces 99.9999% of the *Listeria monocytogenes* in the sous-vide products by reaching 70 °C in the thickest part for 2 min (or its equivalent). This treatment is recommended for the food service and shelf-life industries. The third category of sous-vide products includes cooked products processed to a minimum temperature of 63 °C, determining taste, texture, and appearance acceptance. This applies to both restaurant and home-prepared dishes. The use of lower temperatures significantly prolongs the process time. Reaching higher temperatures for poultry (minced meat, mechanically crushed, injected, and stuffed meat) is recommended for microbiological reasons. In the case of lightly processed sous-vide products, some temperature regimes do not provide fully cooked products, affecting the quality and safety of the food [2,17].

Over the past few years, scientific articles have explored the physical, biochemical, and microbiological properties of SV cooking. These findings have shown that this technique enhances meat consumption quality by improving texture, nutritional properties, and digestibility [18]. In this review, we provide a critical appraisal of sous-vide as a new method of meat cooking, including the types of methods and sous-vide products, the effect of the sous-vide method on meat quality, and the relationship between meat quality and its physicochemical and sensory indicators. The aim is to present the mechanisms of the changes in meat responsible for the desired features of products.

All data presented in this review were summarized from the references, including scientific journals and book chapters. These references were systematically searched against these databases: PubMed, Web of Science, Scopus, and Google Scholar; the following keywords were used: sous-vide, pork, beef, poultry, cooking yield, sensory quality, nutritional value, and microbiological quality. To search for the maximum relative references, the keyword was set as “sous-vide and technological process”, and was restricted to 2000–2020 years.

## 2. The Effect of the Sous-Vide Method on Yield and Water Content in Meat

The sous-vide processing of red meat (beef, pork) and poultry (chicken breast) was carried out by various authors at a wide range of temperatures (50–100 °C) and times (1–72 h) (Table 1). The authors in the publications (Table 1) refer to both the duration of the pasteurization itself (marked as P) and the total duration of the thermal treatment (marked as H), including the time taken to reach the pasteurization temperature. In many publications, there is no information about the process time because it was carried out until a certain temperature was reached in the geometric center of the product. Based on the available data, it can be concluded that the yield of the process decreases with increasing temperature, and time seems to be of rather less importance. Furthermore, the carcasses’ age [19,20], aging time [21], pre-treatment, like tumblering or brining [22], and muscle type [14,19] have an effect on the cooking loss level.

Because high cooking loss leads to low juiciness, yield is an important parameter affecting eating quality [58]. With the sous-vide method, it is challenging to evaluate the yield and water content in meat products compared to traditional methods due to the process parameters and raw material weight and dimensions. Weight loss occurs due to heat-induced protein denaturation and shrinkage, which release a mixture of liquid and soluble matter [59].

The sous-vide method reduces culinary losses due to a lower process temperature, vacuum packaging, and reduced water loss from meat shrinkage and juice loss [38,42,60,61]. In addition, the free acid groups released during cooking can change the pH of the meats. Low-temperature long-time processing (LTLT) is used for red meat, often of poor culinary quality, to increase tenderness but induce heat losses. Sous-vide products typically have a higher yield and water content, but excessive process parameters may lead to equalizing these values with boiled samples.

***Beef*.** The sous-vide processing of beef meat (whole muscle or slices 1.5–4.0 cm thick) was carried out by the authors in the temperature range of 50–100 °C and for 30 to 1080 min (for 18 h). Diverse process parameters result from the meat structure, fat content, and portion size (size and thickness). Beef requires a longer heat treatment process compared to other types of meat, resulting in a lower yield. The yield of the obtained products was affected by the temperature and duration of the process, with temperature being the decisive factor. An increase in either of these parameters resulted in a decrease in the yield of the process (Table 1). Only during sous-vide processing at 50 °C does it achieve 89.2–91.7% yield, similar to chicken processing (Table 1). However, higher temperatures and longer thermal treatments lead to lower yields of 61.1–75.1%. Extending the sous-vide process at a given temperature had a lesser effect on the water content of the product.

A comparison of LTLT (low temperature, long time) and HTST (high temperature, short time) sous-vide showed that HTST (~100 °C by 2 h) resulted in a yield of 64.2%, while the LTLT treatment (75 °C by 36 h) efficiency reached 61.1% [35]. The same results were achieved by other authors [62]. Cooking at higher temperatures for longer periods of time increased the sous-vide cooking loss.

The use of different combinations of temperature (65–95 °C) and time (0–231 min and up to achieve 80 °C inside the product) for cooking slices of beef (*Semitendinosus musculus*) led to the highest yield (75.1%) in 135 min at 65 °C [29]. In the LTLT method, increasing the process temperature from 50 to 65 °C leads to increased mass losses and decreased cutting force, while the effect on the process time remains insignificant [23]. The meat pieces cooked at 50 °C had lower cooking losses compared to 60 °C or 80 °C [22]. The increase in cooking loss was significant from 60 to 70 °C but not from 70 to 80 °C. The muscle fibers show lateral shrinkage and swelling during the initial 6 h of cooking at 60–70 °C and the first two hours at 80 °C, followed by changes independent of water transfers [62]. Meat from older animals required a higher temperature and prolonged cooking to achieve equal tenderness relative to the meat from young animals [19].

***Pork.*** Sous-vide pork was prepared in the range of lower temperatures (48–80 °C) and a process time of 300–1440 min (24 h), as shown in Table 1. A shorter process time of 72 min was used by Kehlet et al. [39]. The highest process yields (84.9–88.2%) were obtained, regardless of the process time, only at temperatures of 48 and 53 °C (Table 1). Higher temperatures led to higher cooking losses, but a longer cooking time at 50 °C reduced them, while the effect was not significant at higher temperatures [38]. The cooking loss of the chops, when prepared with the sous-vide method, increases with the duration of postmortem aging of the meat [21].

Zielbauer et al. [63] reported that the highest weight losses during the sous-vide processing of pork were recorded at temperatures of 60–74 °C. According to these authors, the losses during cooking increase with the set time and temperature, with the highest losses of water occurring in the first 240 min and at temperatures above 60 °C. The meat cooking loss increased with the temperature and heat treatment time, but no correlation was found between loss and moisture content [41].

***Poultry***. Most research on the sous-vide method refers to cooking chicken breast. The authors used temperatures of 55–80 °C and a process time of 35–480 min (Table 1). The high yields of the sous-vide process were 89.4–93.4%, depending on the process parameters, and were reached in the range of 60–65 °C and in under 60 min (Table 1). 

Table 2 juxtaposes the yield and water content of the meat processed using varied heat treatments with various temperatures and process times, reported by many authors. As the process temperature increased up to 100 °C and more, the yield of the meat increased from 79.4 to 65.8% for beef, from 83.3 to 58.7% for pork, and from 93.4 to 71% for poultry (about 13.6–24.6% average). However, the differences in the water content of the raw materials subjected to various methods differed in beef from 70 to 53%, in pork from 68.4 to 58.4%, and in poultry from 74.9 to 53.9% (about 10–17% average).

Model studies by Zhang and Wang [69] assessed the temperature (55, 65, 75, 85, and 100 °C by 30 min) effects on Qingyuan chicken breasts in China, revealing increased losses with higher temperatures (100 °C). Similar observations were made with duck meat [70]. The greatest losses occurred at temperatures of 75–85 °C [68], and in the study of Wattanachant et al. [71] at 80–100 °C. The differences in the results may arise from the thermal process conditions and sample size during the sous-vide treatment.

The temperature and duration of the process affect the yield of poultry cooking. Chicken breast meat processing at 61 °C and a shorter cooking time minimizes the cooking loss. A longer cooking time increases the cooking loss by 0.7–4.8%, with smaller increases observed at 55 °C and 58 °C [45]. Long-term exposure to high temperatures causes water loss, which reduces product weight and yield. Increasing the temperature from 60 to 70 °C increased the cooking losses by 11–12% while increasing the time from 1 h to 3 h increased the losses by 6–8%. The poultry cooking losses range from 24 to 32% due to liquid leakage [48], which contains mainly water and nutrients and changes the chemical composition of the meat [72].

The sous-vide method improves the meat cooking yield by approx. 7% in red meat and 19% in chicken, resulting in less water evaporation compared to conventional thermal treatment methods. This is evidenced by the slightly higher water content (by 2–8%) in sous-vide than in the case of conventional cooking methods (Table 2).

Many authors [44,73] suggest using other thermal processing methods, such as frying or grilling, before the sous-vide method to improve the sensory quality of the products. Madore et al. [73] suggest combining the sous-vide method with frying or grilling to reduce weight loss by 20%. The searing process before the sous-vide treatment enhances the appearance and sensory quality of pork patties. However, longer searing times result in poor cooking properties and color. The optimal searing time is 60 s for improved palatability without compromising storage stability [44].

A comparison of the yields of traditional and sous-vide cooking methods showed that sous-vide allows for a higher cooking efficiency (about +10%) than boiling [48,66,67]. The previous study’s results demonstrated an even four-fold reduction of cooking losses in sous-vide and a 19% efficiency increase when compared to traditional [53].

The cooking loss in meat increases with cooking temperature [74] and is attributed to changes in the meat structure that can explain the changes in the amount of leakage (decreased yield). Muscle fibers contract transversely between 40 °C and 60 °C, increasing the space between them [75]. Water is displaced from the muscle fiber into the interfibrillar space at 46 °C. Actin and myosin denature at 40–50 °C, while myofibrils contract and water moves to the interfibrillar space at 45 °C [76]. Further heating of the meat from 60 to 65 °C causes longitudinal fiber shrinkage, reducing water holding and binding capacity. This shrinkage increases [3] with temperature, and intense leakage is linked to myofibrillar protein denaturation and collagen fiber shrinkage [69,70,77]. Along with water loss, meat also loses substances dissolved in the sarcoplasm of the muscle fibers during cooking. The largest increase occurs between 50 and 70 °C and decreases thereafter [76].

## 3. The Effect of Sous-Vide Heat Treatment on the Sensory Quality of Meat

Biochemical and physical changes occur during the heating process that affect the microbiological quality and sensory characteristics. Many studies indicate better sensory characteristics of beef, pork, and poultry after sous-vide [78,79]. The factor that determines the sensory quality of sous-vide meats is the right combination of time and temperature. Appropriate temperatures in this method make it possible to control fast and slow changes in the products and affect their quality [3]. Vacuum conditions are given a minor role [40,80], but they are important during storage [81,82].

**Color.** Sous-vide-cooked meat is brighter and redder than conventionally cooked meat due to the changes in myoglobin during cooking [32,41,48,57,78,79,82,83].

**Aroma and taste**. Sous-vide cooking improves the shelf life and flavor of the meat [3]. Sous-vide products have an extended shelf life by preventing re-contamination, humidity changes, oxidation, and aroma loss [56]. In contrast to tenderness and juiciness, the flavor of cooked meat (mainly attributed to volatile aromatic compounds) usually develops at temperatures above 70 °C. According to Calkins and Hodgen [84], sous-vide meat cooked at lower temperatures has a worse taste than meat cooked at higher temperatures due to the degree of the Maillard reaction. The specific flavor of cooked meat is determined by the increase in aromatic compounds derived from the breakdown of amino acids [35,85] formed during the thermal degradation of the proteins [86]. Nitrogen rings and derivatives of furfurol, thiophene, and thiolate are formed at 70 °C due to sulfur amino acid changes. Cooked meat’s taste appears even with gentle heat treatment (at 50 °C) and is mainly related to protein hydrolysis and nitrogenous extract conversion. Higher process parameters increase the intensity of the meaty and brothy flavors and odors [79,80,87,88]. At lower temperatures, the process time does not affect the development of the umami (broth) flavor [18]. Higher process parameters reduce the formation of the metallic and off-flavors [35,80], boar taint [89], and “warmed-over” [90] flavors.

**Texture.** Tenderness, juiciness, and taste are the main factors affecting the choice and acceptance of cooked meat by consumers. Low-temperature methods are sought to enhance the sensory features while ensuring product safety for consumption.

Tenderness is a crucial feature of meat quality [91] and is associated with the amount of connective tissue, sarcomere length, and post-mortem protein degradation. The meat microstructure significantly impacts tenderness and hardness [24]. Heat treatment modifies meat structure and ingredients, influencing sensory quality. Denaturation, aggregation, and the degradation of myofibrillar and sarcoplasmic proteins and connective tissue depend on time and temperature during LTLT treatment [4,91].

Meat tenderness is influenced by the cooking temperature and time, with an increased temperature causing stronger fiber shrinkage than a prolonged cooking time. The muscle fibers begin to contract at 35–40 °C and continue until 80 °C. The sous-vide method with lower parameters (50–65 °C) produces tender, juicier, and less hard products, particularly for low-quality meat [40,91,92,93,94].

Protein changes, particularly collagen denaturation and proteolytic activity, are key factors in increasing meat tenderness [91]. In the sous-vide method, meat achieves tenderness through reduced protein denaturation, weakening the connective tissue through collagen solubilization, and water retention. Sous-vide products maintain a consistent temperature throughout their volume, unlike traditional methods that subject the outer layer to high temperatures [95].

As cooking temperatures increase, meat protein denatures. Sarcoplasmic proteins aggregate at 40–60 °C, while myofibrillar proteins unfold at 30–32 °C. Cross-linking occurs between 36 and 40 °C, and gel formation occurs between 45 and 50 °C. Collagen denaturation happens between 53 and 63 °C, and higher temperatures form gelatin [75]. Muscle protein denaturation and fiber reduction start above 60 °C [96], causing meat doneness [76].

Collagen (5.3% of meat protein), a structural protein that contributes to meat palatability and tenderness, is a key component of intramuscular connective tissue (IMCT) in meat [97,98]. As the animals age, the collagen cross-links mature and become more thermally stable, making them more difficult to break. Depending on the thermal properties of the collagen, IMCT affects the tenderness of meat differently at different temperatures and cooking times [15]. The thermal denaturation of collagen is used to improve texture without altering digestibility. IMCT significantly impacts the muscle shear force, but its contribution at 70–80 °C is smaller than the myofibrillar component [98]. Sous-vide cooking has been linked to increased collagen solubility [8].

Heat treatment significantly affects meat tenderization, which occurs through the transformation of collagen at 60–70 °C. Temperature is crucial in reducing collagen-induced hardness. Poultry meat does not need extensive thermal treatment due to its procollagen content, allowing easy swelling in aqueous environments. The denaturation of collagen protein ranges from 53 to 63 °C [74], while the denaturation of myofibrillar proteins (mainly myosin) occurs at 40 to 60 °C. Collagen fiber gels at 60–70 °C, followed by actin denaturation at 70–80 °C [91].

Low-temperature heating converts collagen into gelatin, resulting in the desired meat texture and improved protein digestibility [99]. Low temperatures prevent meat structure changes, fiber shrinkage, and increased juiciness, enhancing protein digestibility and the meat’s flavor.

To sum up, the effect of the sous-vide cooking parameters on the meat texture seems to be beneficial. Generalizing the results is challenging. The texture is influenced by many factors, including muscle cell diameter, collagen thermostability, and the type of meat. These differences can result from factors such as animal age, maturation time, muscle type, storage conditions, and cooking time [37,78,96,100]. The multi-stage sous-vide method of meat preparation, which involves thermal treatment at increasing temperatures in stages, has been increasingly emphasized [65]. LTLT parameters (low temperature, long time) are conducive to better digestibility [39] and obtaining the sensory characteristics corresponding to the needs of older people [25]. However, too long a time and too high a temperature during this process [26,100] negatively affect the juiciness of the meat. For this reason, exogenous enzymes [33,101] are added to reduce the process time, or food additives are added to bind the water expelled from the muscles as a result of thermal shrinkage [32,102,103,104].

**Beef.** In the case of sous-vide beef, research mainly focuses on obtaining the right aroma and texture. The sous-vide method for beef meat reduces the volatile aromatic compounds compared to other thermal processing methods. Research by Rinaldi et al. [35] found a higher aromatic compound content in beef cooked for 2 h at 100 °C compared to the sous-vide samples. The sous-vide method preserves aromas better during storage and reduces unpleasant aftertastes (e.g., hexanal and 3-oktanone).

Clausen et al. [18] found that the content of the volatile compounds in meat depends on the texture of the raw material. Heat-treated products may entrap these compounds in gelling proteins, hindering their release into the headspace. Aldehydes, ketones, and esters are the main substances formed in meat due to lipid oxidation, while aliphatic alcohols and hydrocarbons may be present in lower concentrations. Boiled beef has more aldehydes and alcohols than sous-vide beef, while sous-vide beef has more sulfur compounds. Cooking at 100 °C led to the creation of more ketones, but sous-vide meat had more 4-methylthio-2-butanone, which gives earthy and roast notes. The time of the process (0–8 h) did not affect the glutamate content, responsible for the umami taste [18].

Mortensen et al. [79,88], evaluating the “eye of round” beef steaks prepared in 3, 6, 9, and 12 h at temperatures of 56, 58, and 60 °C, found that the preservation of the sensory features depended on the process parameters. As time and temperature increased, surface browning, boiled veal flavor and odor, brothy odor, and mouth residual intensified. Pink color, juiciness, firmness, veal flavor, and blood/metal flavor decreased [79,88]. The texture characteristics (hardness, gumminess, chewing time) increased with temperature and decreased with a prolonged process time. Meat tenderness increases with the process time and decreases with temperature. Veal and liver odor, liver flavor, and sweet taste were unaffected by time and temperature [79].

Naqvi et al. [19] studied the heat-soluble collagen in young and matured steaks using three combinations of time, temperature (1, 8, 18 h, and 55, 65, and 75 °C), and meat maturation (0 and 13 days). They found more heat-soluble collagen in young animals and lower Warner–Bratzler values as time and temperature increased. This is due to the solubility of the connective tissue in the steaks.

Botinestean et al. [25] found that sous-vide beef steaks (60 °C, 270 min) had higher tenderness, chewiness, and gumminess compared to those cooked at 75 °C and cooked up to 70 °C inside the product. This process reduced meat hardness and affected the cross-section appearance by forming a thin layer of denatured myoglobin, potentially meeting older people’s needs.

Suriaatmaja and Lanier [105] showed a positive effect of the sous-vide method on the tenderization of lower-culinary-quality beef cuts. Sous-vide processing at 50–58 °C for 24–48 h is suitable for tenderizing lean and tough beef. Shear force analysis showed that softening starts at 51.5 °C. Samples treated at 56 °C for 24 h and grilled at 65 °C had similar shear forces to bromelain-softened and grilled samples, indicating a texturizing effect due to collagen degradation. This method involves thermal treatment at rising temperatures, resulting in a slight hydrolysis of myosin and a minor role for the endogenous proteases. The multi-stage sous-vide method (m-SV, 1 h at 39 °C, 1 h at 49 °C, 4 h at 59 °C) reduces the shear forces by 23–27% for raw materials compared to one-temperature methods [14]. Similar results were also obtained by Ismail et al [13].

Tender sous-vide meat results from the increased gelatinization of the connective tissue, as proven by a significant difference in the maximum shear force between oven-roasted (103 N) and sous-vide-cooked meats (76 N) [24]. The LTLT process increases tenderness, solubility, and color but reduces juiciness and causes a brighter color [26].

García-Segovia et al. [93] reached similar conclusions using different sous-vide processing parameters: temperature (60–80 °C), time (15–60 min), and process conditions (atmospheric pressure, sous-vide, and cook-vide). The sous-vide method reduces the shear forces with temperature and process time, softening the connective tissue by dissolving the intramuscular collagen due to the moist environment in the package. The authors pointed out the protective effect of the vacuum packaging on myoglobin, resulting in a redder meat color.

Higher sensory quality compared to traditional methods is also noted by other authors, e.g., beef in Korean sauce [81], beef shank cuts [106], and veal [65].

In conclusion, the sous-vide process allows for very tender textures of hard pieces of meat, improving beef tenderness, which is associated with increasing the hydrolysis of myosin heavy chain, resulting in a higher myofibrillar fragmentation index, collagen solubility, and longer sarcomere length. This is due to the proteolysis of the myofibrillar protein and collagen induced by cathepsin B and L and the limited longitudinal shrinkage [107]. This method also enables excellent control of the degree of doneness [10] and also improves the sensory quality, color, taste, and nutritional value of food [9].

**Pork.** Rotola-Pukkila et al. [86] showed that the concentration of the compounds responsible for the umami taste in sous-vide pork meat and its meat juice depended on the temperature and time of the process. However, the process temperature was more important, which resulted in a higher concentration of amino acids in the meat juice when prepared at 80 °C compared to the samples prepared at 60 and 70 °C. This indicates thermally induced protein hydrolysis, which results in the transfer of free amino acids from the meat into the meat juice. The influence of the parameters on the concentration of inosine-5’-monophosphate (IMP) was not demonstrated; however, a statistically significant relationship was found with the amount of adenosine-5’-monophosphate (AMP).

The creation of volatile aromatic compounds from lipid degradation is positively impacted by temperature and negatively by time. However, volatile compound formation from protein degradation depends on factors such as packaging type, with packaging having a minimal influence. Long process times combined with moderately high temperatures stimulate the formation of volatile compounds with amino acids, resulting in desired meat and roast flavor notes. On the other hand, these parameters reduce the volatile compounds in meat, preventing off-flavors caused by fatty acid degradation [80].

Lepper-Blilie et al. [90] showed that sous-vide cooking and the subsequent storage of roast pork can minimize the undesirable “warmed-over” flavor of the meat. The intensity of the off-flavor of the samples stored for 72 h and then restituted did not differ significantly from those not stored. The SV samples reheated in a bag were characterized by a higher intensity of brothy and fatty flavor than those assessed immediately after roast preparation.

Diaz et al. [82] found that sous-vide pork loin (12 h at 70 °C) deteriorated in sensory quality after ten weeks of storage at 2 °C, with a 45% decline in appearance due to darkening and greying. Storage reduced meat odor and flavor intensity (−59%) and texture features, intensifying a rancid, warmed-over, sour meat flavor and odor. The sous-vide pork suffered sensory deterioration during refrigerated storage, limiting the shelf life of meat-based dishes due to a loss of acceptance (−65%). Sensory deterioration preceded microbiological spoilage.

The sous-vide method can effectively mask negative sensory features in meat, such as boar taint caused by androstenone and skatole accumulation in adipose tissue [89].

Del Pulgar et al. [40] assessed the chemical composition and texture of sous-vide pork cheeks after heat treatment at 60 °C or 80 °C for 5–12 h and various packaging conditions. Traditionally cooked meat texture parameters (hardness, chewiness, and cohesiveness) were similar to the sous-vide samples cooked at 60 °C (5 and 12 h) and 80 °C (5 h) and significantly higher at 80 °C (12 h). Meat cooked at 60 °C has higher adhesiveness than meat cooked at 80 °C. The reason for this is the increased release of water at 80 °C; however, this does not change the texture. The meat texture is influenced by time and temperature combinations, causing muscle connective tissue degradation.

Wang et al. [108] studied the effects of low temperature (55 °C vs. 60 °C) and extended cooking times (4 h, 8 h, 24 h) on the water-holding capacity and tenderness of *Longissmus dorsi* muscle in pork. Increasing LTLT parameters significantly influence pork quality, protein denaturation, cooking losses, and the surface shrinkage coefficient. The perimysium myofibril and collagen fibers contracted, while the surface hydrophobicity increased. Although at higher temperatures the collagen content decreased, its solubility increased. The heating temperature significantly impacts meat quality and protein denaturation, which can be mitigated by increasing the heating time. Significant longitudinal shrinkage coefficient differences were observed at different heating temperatures, possibly due to the denaturation of insoluble collagen [108].

Becker et al. [100] discovered that pork meat tenderness and juiciness depend on the processing temperature, with lower temperatures resulting in juicier meat and higher temperatures resulting in less juicy but more tender meat. The combi steamer can be an LTLT alternative to sous-vide treatment. Becker et al. [100] processed pork for 20 h in a combi steamer (53 °C or 58 °C), comparing it to conventionally prepared meat (180 °C) and the low-temperature LT (60 °C) process. The results showed that the conventional meat was less juicy than LTLT at 53 °C or after reaching 60 °C inside. The LTLT samples at 53 °C and 58 °C were softer than the LT and conventional methods, while the conventional meats were less tender and had the least perceptible metallic taste. The LT and LTLT 53 °C samples showed the highest red color intensity, with LTLT 53 °C being the most acceptable method for juicier, softer meat, and a more intense red color. Previous research by Becker et al. [109] revealed that temperature (53 and 58 °C) is a significant factor affecting meat color and tenderness.

Jin et al. [94] compared sous-vide with other heat treatment and maturation methods using texture parameters in ham and pork tenderloin. Sous-vide cooking (12 h at 70 °C) can decrease the shear force, hardness, and chewiness of the silverside to values comparable to or lower than those reported in grilled and roasted loin. Sous vide exhibited superior color, texture, sensory attributes, and oxidation stability in the pork samples compared to the boiled and steamed samples [110].

**Poultry.** Kurp et al. [41] found that sous-vide cooking at 60–65 °C for 4 h is the most sensory-acceptable, with tenderness and juiciness being the most important sensory features. Other authors [67,111] also found similar relationships. Researchers [66,67,112] indicate a better sensory quality (aroma, appearance, color, flavor, texture, tenderness, juiciness, and overall quality) of sous-vide chicken compared to conventionally cooked meat, as well as good maintenance of this quality after storage [113].

The hardness and intensity of the cooked meat taste increased with an increase in the process parameters, while the juiciness and aftertaste intensity decreased [66]. Sous-vide meat maintains uniform juiciness across its entire cross-section [112]. In the other study [53], sous-vide-cooked poultry breast had higher scores for taste, smell, color, and general appearance and was more tender and juicier than when cooked in water. In the study of Karpińska-Tymoszczyk et al. [45], a chicken breast fillet achieved the desired sensory features at lower temperatures (55–61 °C), having the highest overall notes when processed for 200 min at 58 °C. Sous-vide cooking enhances the sensory quality of normal and white-striping chicken breasts [49].

Biyikli et al. [114] reported that sous-vide cooking (at 65, 70, and 75 °C for 20, 40, and 60 min) decreased the turkey chop yield, moisture content, and elasticity with an increasing process temperature. However, the cooking losses, fat content, pH, hardness, cohesiveness, gumminess, and chewiness increased. Consumers rated turkey chops cooked at lower temperatures higher, with process temperature having the greatest influence on the poultry meat changes.

According to various authors [3,95], poultry meat remains juicy and tender when processed at 60–65 °C in the core, while higher temperatures are recommended for meats with higher fat content. Sous-vide cooking provides sufficient thermal denaturation of intramuscular connective tissue (55–60 °C), improving chicken tenderness. However, elevated temperatures may eliminate the positive effect on the water-holding capacity.

Sous vide significantly affects the water-holding capacity, cooking losses, protein solubility, shear force, myofibrillar fragmentation index, and lipid oxidation. Chicken breast and leg were cooked sous vide (at 55 °C and 65 °C for 3–6 h) compared to conventionally cooked meat at 75 °C (core temperature 71 °C). No significant differences in pH or color brightness were observed between the sous-vide cooking conditions. Sous-vide cooking at 65 °C reduced the shear force and collagen content, possibly due to collagen denaturation. Sous-vide cooking at 55 °C reduced weight loss and preserved the water content of chicken breast and thigh [115]. Sous-vide cooking of duck breast reduced weight loss compared to weight loss in conventional methods (cooking, frying, and baking) while enhancing the meat’s gumminess, chewiness, and elasticity. Cooking duck breast at 65 °C for 1.5 h may be optimal for reducing weight and improving texture. Breast stored at 4 °C remains safe with no microorganisms and unchanged physicochemical properties within 7 days [116].

## 4. The Effect of Sous-Vide Heat Treatment on the Nutritional Value of Meat

Table 3 shows sous-vide technology’s impact on the nutritional value of meat dishes such as poultry, pork, and beef compared to conventional methods. 

A compilation of the results from various studies (Table 3) shows a higher protein content in beef and poultry prepared sous vide compared to raw meat. The sous-vide method has a less profound effect on raw materials compared to grilling and roasting. Frying increased the fat content of beef and poultry while roasting and grilling had limited increases in protein and fat content compared to raw meat. It is problematic to draw summarizing conclusions on proximate composition due to the limited number of studies conducted and the differentiated process parameters. These findings thus need to be interpreted with caution. 

Few authors have analyzed sous-vide processing’s impact on protein, amino acid, fat, and fatty acid transformation.

The share of protein and fat in cooked meat increases by more than 50%, while the minerals and water contents decrease and are exuded as meat juice. Cooking meat sous vide offers increased nutritional value due to concentrated nutrients and better retention of vitamins, minerals, and unsaturated fatty acids [8,27,64]. However, it also transfers nutrients to the decoction, which is a potential disadvantage [87].

Ramane et al. [56] reported that the sous-vide cooking of poultry fillets causes a slight loss of soluble proteins, while the losses are twice as high in meat cooked with vegetables and fruits compared to plain meat. 

Silva et al. [54] found no significant differences between desalted jerky poultry meat and the sous-vide samples (8 h at 65 °C). The sous-vide samples had the fewest free thiol groups, suggesting that sulfur amino acids are less resistant to prolonged heating than aromatic amino acids such as tryptophan. They suggest that the extended heating time affects the destruction of the free thiol groups, the formation of Schiff bases, and meat tenderization. Temperature is related to protein degradation, carbonylation, and disulfide bridge formation. Increased heat treatment parameters increased the conjugated dienes and decreased TBARS and hexanal content. Protein carbonyl groups increase with time at 60–80 °C. Treatment in this temperature range leads to increased aminoadipic (AAS) and glutamine (GGS) semi-aldehydes content and a significant breakdown of secondary compounds during lipid oxidation.

Sous-vide heating reduces lipid oxidation, as indicated by the TBA value, rather than fatty acid composition [118]. Roldan et al. [119] found that the oxidative changes in the proteins and fat affect meat taste and texture. Ramane et al. [56] revealed that sous-vide heat treatment reduced amino acid degradation by 3.5% and increased methionine losses.

Heating meat seems to increase protein digestibility, depending on the heat transfer and heating rate [75]. Controlling these variables determines the meat’s palatability after cooking.

Meat proteins are categorized into myofibrillar (50–55%), sarcoplasmic, and connective tissue proteins, with varied digestibility depending on function and location [120,121]. Protein is highly sensitive to heat treatment, undergoing modifications like aggregation, cross-linking, oxidation, conformational changes, and reduced solubility, resulting in protein structure changes. Meat proteins form gel networks or partially developed structures based on the temperature and cooking time [121]. These changes affect not only the physicochemical and sensory properties of the meat but also its digestibility, which may impact nutrient absorption. Heat treatment induces oxidative modifications in proteins, impacting myofibrillar protein sensitivity to proteases. The digestive speed depends on the digestive enzyme, cooking time, and temperature. Cooking affects myofibrillar protein digestion, while connective tissue protein surrounds the muscle fibers and plays an auxiliary role in meat. No studies have focused on intramuscular connective tissue digestion behavior.

Kehlet et al. [39] examined the impact of the cooking methods, time, and pork structure on appetite regulation and in vitro protein digestibility. No significant differences were found in the subjective appetite assessment or hunger feeling 3 h post-consumption. Nevertheless, LTLT cooking at 58 °C for 72 min appeared to improve in vitro protein digestibility. 

The assumption referred to the research by Christensen et al. [37], who found that sous-vide LTLT pork cooked at 58 °C had higher proteolytic activity from endogenous cathepsin B+L than pork cooked at higher parameters. However, higher peptide and free amino acid contents were not observed in the LTLT samples. The in vitro digestion results show the thermal treatment type and time impacted the gastric proteolysis rate, while raw material comminution did not. LTLT cooking for 72 min increased the proteolytic rate in the early gastric phase, compared to 17-h or oven cooking, without affecting the intestinal phase [39].

Falowo et al. [30], who studied the effect of the sous-vide method on cow fatty acid profiles and minerals in meat and liver, found no differences in their total content based on variations in the sous-vide method parameters (120 min at 65 °C and 60 min at 85 °C). The meat treated at 85 °C had a higher total content of PUFA, n-6, PUFA/SFA, and PUFA/MUFA ratios compared to the meat processed at 65 °C. The sous-vide treatment reduced the n-6 fatty acid proportion and PUFA sum [122]. Rasinska et al. [123] found increased Ʃ SFA and decreased Ʃ PUFA in SV rabbit meat compared to raw meat, with levels lower than those observed in roasted meat. The low parameters of the sous-vide process used in the Gluchowski et al. [66] study resulted in the least changed profile of fatty acids in the meat compared to raw meat.

The results of Wereńska et al. [124] showed that heat treatment increases the Ʃ PUFA n-6/n-3 ratio, but sous vide had the lowest value. Sous-vide cooking is more beneficial for consumers in terms of Ʃ DFA/Ʃ OFA, Ʃ UFA/Ʃ SFA, NVI, health-promoting indexes (HPI), and inflammatory biomarker indexes compared to other methods used, especially in the case of meat with skin. However, stewed meat without skin was more favorable than SV and microwave cooking in terms of various lipid indicators.

There are too few articles on the influence of sous-vide processing on macronutrients, minerals, and vitamin content to draw clear conclusions about the changes in meat after sous-vide heat treatment. Based on scarce data, it can be concluded that lower process parameters are conducive to lower losses of B vitamins [35], amino acids [56,86], and lower HAA content [34,124]. Additionally, the retention of minerals [30,125,126] is at a similar or higher level than in boiled samples. When using higher parameters, similar to conventional methods, this effect was not determined [127,128]. 

The sous-vide processing of beef meat and liver at 85 °C reduces K and Mg content in meat and increases their content in the liver compared to raw material. However, lowering the process temperature to 65 °C did not alter the Zn, Ca, Mg, and K content of beef meat and liver [30].

Data by da Silva et al. [127] suggest that the sous-vide process increases mineral bioavailability. The authors found an increase in Ca bioavailability of 33.7%, Cu of 12%, Fe of 28.3%, Mg of 12.9%, and Zn of 18.3% in sous-vide beef liver compared to boiling. However, the percentage of bioavailable potassium remained constant at 42.6–43.9%. The differences may result from greater losses of minerals during boiling related to their exudation into the medium.

Research by Macharáčková et al. [126] indicates that sous-vide pork cooked at 70 °C maintains higher mineral concentrations than methods such as broiling, enhancing sensory properties and providing higher biological value for consumers. Sous-vide and steam-cooked beef meats are good sources of Fe and Zn. Sous vide exhibits higher retention values for K and Na and lower retention for Cu, Fe, and Zn when compared to steamed pork [129].

Barnett et al. [130] assessed the effect of the sous-vide method (80 °C, 6 h) and steak frying on the level of minerals (Mg, Al, Ca, V, Mn, Fe, Co, Cu, Zn, Se, and Br) in the blood of 12 healthy men. According to the authors [130], the consumption of a 270 g beef steak altered the plasma Fe and Zn levels, and the cooking method (PF vs. SV) may have influenced the changes in circulating Zn. However, no significant differences were found in the case of the other minerals.

No significant differences between sous-vide and traditional pork tenderloin in vitamin B1, B2, and B6 losses during the preparation day were determined, as the same parameters were used. Storing the roast for 3 to 14 days generated further losses of vitamins B1, B2, and B6, ranging from 42.5 to 45.6% and 12–51%, respectively [128]. Rinaldi et al. [35] found similar relationships when assessing vitamin B3 and B12 losses in sous-vide beef under different processing conditions (2 h at 100 °C, 36 h at 75 °C). Sous-vide beef cooked at 100 °C had a 70% loss in vitamin B12 content, which was comparable to traditional cooking methods (80%) and higher than at 75 °C (50%). This treatment caused the vitamin B3 losses to be significantly lower in the sous-vide samples (60–65%) compared to the water-boiled samples (78%), likely due to the limited elution to water.

Intensive traditional cooking causes quality deterioration and heterocyclic amine formation due to overcooking. Lower heat treatment temperatures are not conducive to heat-induced harmful substance generation. Oz and Zikirov [34] found that sous-vide cooking reduces the formation of heterocyclic aromatic amines (HCA) in beef compared to frying. The total HCA content in the sous-vide-cooked samples was significantly lower than in the pan-fried samples and increased with the process time, regardless of the temperature applied.

The sous-vide samples cooked at 75 °C for 2 h had 15.6% lower total heterocyclic amine contents compared to the traditionally braised samples but were significantly higher than the sous-vide samples at 65 °C for 2–3 h. Lower temperatures hinder HA generation, and SV cannot effectively reduce the residual HA content for an extended duration. High HA content may infiltrate from reused marinades [68].

## 5. The Effect of Sous-Vide Heat Treatment on the Microbiological Quality of Meat

A review of the studies on the microbiological quality in sous-vide meat (Table 4) shows that this method allows for a higher microbiological quality of stored samples than conventional methods, depending on the raw material type, initial microbiological load, and adequacy of the applied parameters.

In the sous-vide method, strict food hygiene rules, high-quality raw materials, and precise control of the process parameters are crucial for the preparation and storage of sous-vide meat [132]. Vacuum-packed products should be stored below 3 °C to prevent *Clostridium botulinum* growth [133]. The safe minimum internal temperature for poultry is 60 °C and 55 °C for other meats. Thermal treatment at lower temperatures must not exceed 4 h; otherwise, the product should be disposed of. The recommended cooling time for products below 3 °C is 2 h, while hygienic standards recommend cooling finished products to 4 °C within 6 h. Packed raw materials can be stored for two days, while cooked products can be stored for three days. The maximum storage time of a fully pasteurized product is seven days [133]. The lower cooking temperatures destroy only the vegetative microorganisms, but packaging can be unsealed. Anaerobic microorganisms, such as *Clostridium perfringens, Clostridium botulinum type E*, and *Bacillus cereus*, as well as facultative anaerobes, such as *Listeria monocytogenes* and *Escherichia coli*, are likely to develop during storage. The strict control of the process regimes and rapid cooling prevent microorganism development during storage [134].

The recent scientific literature highlights the significant risk of developing microbiological hazards related to the consumption of sous-vide dishes. In 2014, three cases of *Salmonella enteritidis* infection were linked to sous-vide food in a restaurant in Canada. In two cases, the infection occurred in the same restaurant at different time intervals after the consumption of a breakfast menu containing, among other things, eggs heated for 2 h in a water bath at 62.5 °C. The third case concerned the consumption of sous-vide duck cooked for 25 min at 62.5 °C. Although other high-risk foods were also served in both facilities, due to large irregularities in maintaining temperature regimes, sous-vide was considered a probable source of infection [135].

Jørgensen et al. [136] evaluated 356 ready-to-eat food samples from caterers (92%), retail (7%), and producers (1%). Only 10% of them described the method as sous-vide, with 29% of the samples being microbiologically unsatisfactory among lightly processed foods. The combination of the sous-vide method with other thermal treatment methods, such as frying and grilling, significantly improved the microbiological quality of the tested samples. *Salmonella* was isolated from five sous-vide duck breast samples and L. monocytogenes from foie gras confit ballotine. The temperature range used in the sous-vide samples ranged from 44 to >100 °C, with the process time oscillating between 10 and 50 min for the samples heated at 50 °C.

Purslow [74] emphasizes caution when using the LTLT method for pork dishes, as it may transmit *Trichinella spiralis* and *Toxoplasma gondii*.

In recent years, the number of sous-vide meals cooked at low temperatures, traditionally considered a “temperature danger zone”, has increased. Its influence on bacterial behavior has not been well studied, which complicates safety assessment [137]. Low cooking parameters and the incompetent application of sous-vide may hinder pathogenic organisms’ inactivation and trigger their multiplication. Special prognostic models have been developed to estimate the survival of *Salmonella* spp. [138] and *Clostridium perfringens* spores [139] in sous-vide beef, *Listeria monocytogenes* [140], and *Listeria monocytogenes* and *Salmonella* spp. [141] in LTLT-treated chicken breast and various animal products [142]. Stringer et al. [2] discussed the issues with using the ComBase system for estimating vegetative pathogen growth and survival at 40–60 °C processing.

Stringer et al. [2] evaluated the heat dose adequacy using 979 recipes from the internet, cookbooks, equipment manufacturers, chefs, and government officials. The findings indicate that most studied parameters for pork tenderloin and beef steaks were too low, but they were often additionally fried and grilled. Many of the too-low parameters were suggested for beef, lamb, and veal, and only minimally for pork. The recommended processing temperature range in recipes for poultry was 55–80 °C. The sous-vide method provides convenient, ready-to-eat food with a long shelf life, e.g., a turkey cutlet (65 °C for 40 min) with a 28-day storage life at 4 °C and 15 days at 12 °C. *Salmonella* spp., L. monocytogenes, and Cl. perfringens were not detected in the cutlet samples during storage [131].

## 6. Conclusions

The literature indicates that the sous-vide (SV) method of preparing beef, pork, and poultry has many advantages, including a higher process yield reflecting in higher juiciness. Due to the prevention of the loss of volatile substances and water, as well as the inhibition of the aftertaste of the oxidation of proteins and fats during heating, the flavor and aroma of the meats prepared by this method are preferred more. These changes also improve the color, tenderness, and juiciness of the meat. In addition, limiting the growth of aerobic bacteria, meat prepared with the sous-vide method and stored in refrigerated conditions also has a beneficial effect on the sensory characteristics of the meat. This method, depending on the process parameters (temperature and time), can also reduce the loss of nutrients (B vitamins and minerals) compared to conventional methods, i.e., boiling. To ensure a better palatability of the meat prepared using SV, depending on the formation of the Maillard compounds, grilling, frying, or roasting can be used additionally before or after SV treatment. 

In all the analyzed studies, the authors state that the yield of the process depends on the type and properties of the meat and decreases with increasing temperature, while the cooking time is not significant. Similarly, the vacuum conditions play a secondary role but affect the shelf life of SV-cooked meat. Beef requires a longer cooking process compared to other types of meat, resulting in a lower yield. Many studies have shown that cooking SV at 50 °C for extended periods resulted in the lowest cooking losses. And these losses, resulting mainly from the changes in the structure and properties of the meat proteins, are the highest at temperatures above 60 °C. A higher process yield results in higher juiciness.

Due to the insufficient amount of data available in the literature, future research should focus on the nutritional value of sous-vide products and their health-promoting properties. Meat products produced using this method, due to the extension of the shelf life, while maintaining sensory quality, can be used both in consumer nutrition in individual households, especially one-person households, as ready-to-eat. They can also become a good solution in the catering sector, where they would contribute to reducing overproduction and food waste.

## Figures and Tables

**Table 1 foods-12-03110-t001:** Effect of temperature and time of sous-vide processing on yield and water content in meat.

Type of Meat	Temp.	Size (cm)	Process Time	Yield (%)	Water Content (%)	References
	(°C)	(Weight (g))	(min)			
Beef	50	W/M	90/390 ^P^	91.7/89.2	-	Vaudagna et al. [23]
	55	(170 g)	60/480/1080 ^H^	84.3/77.3/74.4	-	Naqvi et al. [19] **
	59	3.0 cm	240 ^H^	77.7	-	Uttaro et al. [14] **
	60	-	60 ^H^	81	-	James and Yang [24]
		2.5 cm	270 ^H^	64.1	-	Botinestean [25]
		4.0 cm (70 g)	1440/2880/4320 ^H^	79.2/80.5/79.4	-	Alahakoon et al. [26]
		2.0 cm (15–20 g)	30/240 ^H^	94.5/67–78	-	Chotigavin et al. [16]
		-	270/600 ^H^	79.1/71.3	-	Bhat et al. [27]
		6.0 cm	720/1440/2160 *	87.2/84.1/81.9	-	Karki et al. [28]
	65	W/M	90 ^P^	80.6	-	Vaudagna et al. [23]
		2.0 cm (160 g)	135 ^P^	75.1	-	Moraes and Rodrigues [29]
		40–70 g	120 ^H^	-	67.8	Falowo et al. [30]
		4.0 cm (70 g)	1440/2880/4320 ^H^	77.7/75.9/78.0	-	Alahakoon et al. [26]
		(250 g)	60 ^H^/480 ^H^	75.6/67.7	67.5/63.9	Naqvi et al. [31]
		(170 g)	60/480/1080 ^H^	73.6/67.1/64.0	-	Naqvi et al. [19] **
		6.0 cm	720/1440/2160 *	84.4/78.9/78.7	-	Karki et al. [28]
	69.4	2.0 cm (160 g)	39/231 ^P^	73.4/67.0	-	Moraes and Rodrigues [29]
	70	C/M	2:00 PM	74.3	-	Szerman et al. [32]
		200 g	30 ^H^	76.3	-	Zhu et al. [33]
		4.0 cm (70 g)	1440/2880/4320 ^H^	73.6/75.7/76.2	-	Alahakoon et al. [26]
		6.0 cm	720/1440/2160 *	81.7/76.9/75.2	-	Karki et al. [28]
	75	2.5 cm	→70 °C ^H^	66.1	-	Botinestean [25]
		3.0 cm	120/240 ^H^	-	61.2/60.6	Oz and Zikirov [34]
		4.0 cm	2160 ^H^	61.1	-	Rinaldi et al. [35]
		(170 g)	60/480/1080 ^H^	64.3/60.0/59.3	-	Naqvi et al. [19] **
	80	2.0 cm (160 g)	→80 °C ^P^	71.6	-	Moraes and Rodrigues [29]
			135/270 ^P^	64.4–69.7/66.3	-	
		1.5 cm (270 g)	360 ^H^	-	55	Nuora et al. [36]
	85	(40–70 g)	60 ^H^	-	59.7	Falowo et al. [30]
		3.0 cm	120/240 ^H^	-	60.2/59.2	Oz and Zikirov [34]
	90.6	2.0 cm (160 g)	39/231 ^P^	62.8/61.0	-	Moraes and Rodrigues [29]
	95	2.0 cm (160 g)	135 ^P^	60.7	-	Moraes and Rodrigues [29]
		3.0 cm	120/240 ^H^	-	58.1/59.1	Oz and Zikirov [34]
	100	4.0 cm	120 ^H^	64.2	-	Rinaldi et al. [35]
Pork	48	4.0 cm	300/1020 ^P^/→48 °C ^H^ *	88.2/85.0/86.8	-	Christensen et al. [37]
	50	1.0 cm	720/1440 ^H^	85.5/88.4	-	Hwang et al. [38]
	53	4.0 cm	300/1020 ^P^/	89.7/84.9/86.8	-	Christensen et al. [37]
			→53 °C ^H^ *			
	55	1.0 cm	720/1440 ^H^	78.9/80.2	-	Hwang et al. [38]
	58	4.0 cm	300/1020 ^P^/→ 58 °C ^H,^*	76.7/72.4/79.4	-	Christensen et al. [37]
		W/M (344.3g)	72/1020 ^P^	78.4/74.7	-	Kehlet et al. [39]
	60	3.2 cm (72.8 g)	300/720 ^H^	82.8/79.8	64.8/66.5	del Pulgar et al. [40]
	60	2.5 cm	120/180/240 ^H^	81.8/78.9/77.6	68.9/67.5/67.5	Kurp et al. [41]
	60	1.0 cm	720/1440 ^H^	78.3/76.5	-	Hwang et al. [38]
	61	2.0 cm (100 g)	45 ^P^	80.1	-	Jeong et al. [42]
	63	4.0 cm	300/1020 ^P^/→63 °C ^H,^*	72.8/69.2/79.4	-	Christensen et al. [37]
		4.0 cm (170 g)	180 ^H^	85.3	64.6	Latoch et al. [43]
	65	2.5 cm	120/180/240 ^H^	80.5/81.6/70.4	68.2/66.9/66.3	Kurp et al. [41]
	70		60/90/120 ^H^	74.8/71.2/70.4	65.9/64.0/65.3	
	75		60/90/120 ^H^	71.0/68.3/63.3	64.9/63.2/62.3	
		1.5 cm (60 g)	120 ^H^	81.6	-	Cho et al. [44]
	80	3.2 cm (72.8 g)	300/720 ^H^	63.9/59.1	58.4/60.8	del Pulgar et al. [40]
Chicken breast	55	2.5 cm (250 g)	260/320 ^H^	88.2/87.4	-	Karpińska-Tymoszczyk et al. [45]
	58		140/200 ^H^	90/88	-	
	60	2.0 cm (130 g)	120/180 ^H^	86.6/83.7	71.2/68.9	Hasani et al. [46]
		2.5 cm (125 g)	60/90/120/150 ^H^	89.8/89/87.5/87.4	71.4/71.3/71.7/71.8	Haghighi et al. [47]
		W/M	60/120/180 ^H^	93.4/88.3/86.8	-	Park et al. [48]
		W/M	120 ^H^	86.7	-	Lee et al. [49]
		W/M	60/480 ^H^	90.5/79.6	-	Kerdpiboon et al. [50]
	60–61	1.5 cm	35 ^P^	89.8/87.6	71.8/71.4	Hong et al. [51,52]
	61	2.5 cm (250 g)	90/150 ^H^	90.7/85.9	-	Karpińska-Tymoszczyk et al. [45]
	63–65	-	60 ^H^	93.4	74.9	Soletska and Krasota [53]
	65	3.0 cm	480 ^H^	-	57.2	Silva et al. [54]
	66	W/M	60/120/180 ^H^	82.7/77.6/74.7	-	Park et al. [48]
	70	2.5 cm (125 g)	60/90/120/150 ^H^	86.0/83.1/81.6/81.3	71.8/71.4/70.0/70.5	Haghighi et al. [47]
	77	W/M	60 ^P^	67	-	Altmann et al. [55]
	80	2.5–3.5 cm (130 g)	40 ^H^	-	69.5/67	Ramane et al. [56,57]
		2.5 cm (125 g)	60/90/120/150 ^H^	82.1/78.2/77.2/75.7	70.4/69.8/69.4/69.0	Haghighi et al. [47]

*—Until the core temperature is reached; ^H^—total time of thermal treatment; ^P^—only pasteurization time; W/M—whole muscles; **—the average means for sous-vide, regardless of other study factors; and →—process was performed to reach given temperature in a product core.

**Table 2 foods-12-03110-t002:** Comparison of sous-vide and other heat treatment methods in terms of yield and water content.

Type of Meat	Heat Treatment Method	Temperature (°C)	Time (min)	Yield (%)	Water Content (%)	References
Beef	sous vide	75	120/240 ^H^	-	61.2/60.6	Oz and Zikirov [34]
		85	120^/^240 ^H^	-	60.2/59.2	
		95	120/240 ^H^	-	58.1/59.1	
	boiling	~100	42 ^H^	-	53	
	sous vide	80	360 ^H^	-	55	Nuora et al. [36]
	frying	240	5 ^H^	-	60.4	
	sous vide	60	240 ^H^	70	-	Modzelewska-Kapituła et al. [64]
	steaming	100	→75 °C ^H^	65.8	-	
	sous vide	60	270 ^H^	79.1	-	Bhat et al. [27]
	stewing	80	45 ^H^	70.2	-	
	sous vide	65	150 ^H^	79.4	70	Kaliniak-Dziura et al. [65]
	grilling	240	→72 °C	73.4	67.2	
	steaming	100	→72 °C	73.3	67	
Pork	sous vide	58	72/1020 ^P^	78.4/72.4	-	Kehlet et al. [39]
	roasting	160	→58 °C ^H^	71.5	-	
	sous vide	60	300/720 ^H^	83.3/79.8	66.5/64.8	del Pulgar et al. [40]
		80	300/720 ^H^	63.9/59.1	60.8/58.4	
	boiling	60	300/720 ^H^	82.8/79.8	66/64.7	
		80	300/720 ^H^	65.8/61.8	60/58.7	
		~100	30 ^H^	63.6	58.9	
	sous vide	61 ^V1^	45/90 ^P^	79.7/78.9	68.4/67.5	Jeong et al. [42]
		61 ^V2^	45/90 ^P^	78.0/78.6	66.0/65.0	
		71 ^V1^	45/90 ^P^	71.4/71,1	64.3/61.8	
		71 ^V2^	45/90 ^P^	68.9/67.1	62.8/61.7	
	boiling	~100	45 ^H^	63	61.4	
	sous vide	71	120 ^H^	73.5	-	Wang et al. [15]
	braising	<100	120 ^H^	58.7	-	
Chicken breast	sous vide	64	60 ^H^	93.4	74.9	Soletska and Krasota [53]
	boiling	~100	30 ^H^	74.2	66.3	
	sous vide	65	480 ^H^	-	57.2	Silva et al. [54]
	frying	170 ± 10	10 ^H^	-	53.9	
	grilling	190 ± 10	10 ^H^	-	55.9	
	roasting	200	15 ^H^	-	54.4	
	sous vide	64	60 ^H^	89.4	74.5	Głuchowski et al. [66]
		66	80 ^H^	82.4	71.3	
		75	35 ^H^	83.1	70.5	
	steaming	100	55 ^H^	72.4	71.2	
	boiling	100	42 ^H^	69.5	68.5	
	sous vide *	70	60 ^H^	82.7	-	Park et al. [48]
	boiling	180 °C	→71 °C ^H^	73.5	-	
	sous vide	76	60 ^H^	88.5	-	Przybylski et al. [67]
	boiling	100	→ 76 °C ^H^	71	-	
	sous vide *	80	60 ^H^	82.1	70.4	Haghighi et al. [47]
	boiling	100	60 ^H^	71.9	68.3	
	sous vide *	75	120 ^H^	82.1	67.5	Cheng et al. [68]
	braising	90	60 ^H^	71.4	65.2	

^H^—Total time of thermal treatment; ^P^—only pasteurization time; ^V1^—vacuum level = 98.81; ^V2^—vacuum level = 96.58. *—for comparison, only the highest temperature and shortest time combination was taken; and →—processes were performed to reach given temperature in a product core.

**Table 3 foods-12-03110-t003:** Effect of sous-vide heat treatment on the content of selected meat components.

Type of Meat	Thermal Treatment Method	Temperature (°C)	Time (min)	Average Content of (%)			References
				Protein	Fat	Ash	
Beef	raw	-	0	-	3.2	-	Falowo et al. [30]
	sous vide	65	120 ^H^	-	3.6	-	
	sous vide	85	60 ^H^	-	3.6	-	
	raw	-	0	21.8	4.6	0.9	Nuora et al. [36]
	sous vide	80	360 ^H^	35.1	6.9	-	
	fried	240	5 ^H^	27.8	12.8	-	
	raw	-	0	22.5	0.7	1.2	Kaliniak-Dziura et al. [65]
	grilling	240	→72 °C	29.5	1	1.4	
	sous vide	65	150 ^H^	26.8	0.9	1.3	
	steaming	100	→72 °C	28.8	1.2	1.2	
Pork	stewing	110	150 ^H^	21.5	13.9	-	Jiang et al. [117]
	sous vide	75	720 ^H^	20.1	4.9	-	
	sous vide	65	480 ^H^	30.2	10.5	-	
Poultry	raw (broiler)	-	0	22.9	1.7	1.2	Ramane et al. [56,57]
	sous vide	80	40 ^H^	26	1.8	2.3	
	raw (hen)	-	0	25.1	1.6	1.2	
	sous vide	80	40 ^H^	28.7	1.8	2.1	
	raw *	-	0	28.2	2.3	6	Silva et al. [54]
	sous vide	65	480 ^H^	35.3	2.8	4.4	
	fried	160–180	10 ^H^	32.8	4	6.7	
	grilled	180–200	10 ^H^	34.6	2.5	6.2	
	roasted	200	15 ^H^	36.4	2.3	6.8	
	raw	-	0	22.6	0.5	-	Głuchowski et al. [66]
	sous vide	64	60 ^H^	22.9	1.9	-	
		66	80 ^H^	27.4	1.2	-	
		75	35 ^H^	24.5	1.9	-	
	steaming	100	55 ^H^	27.4	1.5	-	
	boiling	100	42 ^H^	29.2	1.4	0	

^H^—Total heat treatment time; *—broiler ’charqui’; and →—processes were performed to reach given temperature in a product core.

**Table 4 foods-12-03110-t004:** Effect of sous-vide heat treatment on the microbiological quality of meat.

Raw Material	Methods and Parameters of Heat Treatment		Microbiological Quality		References
			on Preparation Day	on the Last Day of Storage at 2–4 °C	
Chicken breast sauté	sous vide	60 min at 63–65 °C	Total microbial count of mesophilic and relatively anaerobic 2.5 log CFU/g	6th day: Total microbial count of mesophilic and relatively anaerobic 2.6 log CFU/g	Soletska and Krasota [53]
	cooking in water	30 min at 95–98 °C	Total microbial count of mesophilic and relatively anaerobic 2.8 log CFU/g	6th day: Total microbial count of mesophilic and relatively anaerobic 3.1 log CFU/g	
	sous vide	up to achieve temp. 35 min at 61 °C	Total microbial count 4 log CFU/g; coli group of bacteria: 2.6 log jtk/g	14th day: Total microbial count 3.0 log CFU/g	Hong et al. [51]
	sous vide	64 °C, 60 min; 66 °C, 80 min; 75 °C, 35 min	TVC in a raw chicken breast (0 days) was 3.9 log CFU/g; yeast and mold counts were 3.5 log CFU/g; coagulase-positive *Staphylococcus, E. coli*, *Listeria monocytogenes, Enterobacteriaceae*, and *Salmonella* spp. were not detected	After 5 and 10 of storage coagulase-positive *Staphylococcus*, *E. coli*, *Listeria monocytogenes*, *Enterobacteriaceae,* and *Salmonella* spp. were not detected. Total aerobic bacteria count: <10^5^ CFU/g	Głuchowski et al. [66]
Spiced turkey breast meat (called turkey cutlet)	sous vide	65, 70, 75 °C × 20, 40, 60 min	After process, total mesophilic aerobic bacteria decreased approx. by 2 log CFU/g for all samples. *Listeria* spp. was detected only in meat cooked at 65 °C for 20 min	---	Biyikli et al. [114]
Turkey cutlet	sous vide	65 °C for 40 min (chilling in 3 °C for 10 min	Total microbial count of mesophilic 2.05 log CFU/g	5.89 log CFU/g after 35 days of storage in 4 °C and after 21 days of storage in 12 °C	Akoğlu et al. [131]
Pork tenderloin sauté	combined method: pre-grilled + sous vide	5 min in 300 °C and 2 h in 70 °C	<1.0 log CFU/g of psychrophilic or relatively psychrophilic microorganisms from the family Enterobacteriaceae and lactic acid bacteria; 2–3.0 log CFU/g yeasts and molds	10th week: <1 log CFU/g of psychrophilic or relatively psychrophilic microorganisms from the family *Enterobacteriaceae* and lactic acid bacteria; 1.90 log CFU/g yeasts and molds	Diaz et al. [82]
Pork loin	sous vide	60 °C and 65 °C for 2 h, 3 h, and 4 h, and at 70 °C and 75 °C for 1 h, 1.5 h, and 2 h.	Total microbial count of anaerobic aerobic microorganisms was reduced after cooking by about 2 logs and ranged from 1.10 to 2.01 log CFU/g. *Salmonella* spp. and *Listeria monocytogenes* have not been detected.	---	Kurp et al. [41]

TVC—Total viable aerobic count.

## Data Availability

The data is contained within the article. The data used to support the findings of this study can be made available by the corresponding author upon request.

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
