# Peer review of "Sous-Vide as an Alternative Method of Cooking to Improve the Quality of Meat: A Review"

_foods, 2023, doi:10.3390/foods12163110_

Round 1
Reviewer 1 Report
The Article entitled „Sous-vide as an alternative method of cooking to improve the 2 quality of meat: a review” is a review paper considering mechanisms of changes in meat and quality parameters in sous-vide products. It is presented in detail, but there is a need to be checked thoroughly by the authors and removes some sentences - repetition, e.g. introduction, row 27-33, 34-37 etc. Literature data are well and critically presented in this review. There may be a more precise conclusion defined and under the aim of the review. It is not a systematic review and meta-analysis paper, but still, authors should point out the method of collecting/searching string, data presented in tables (1, 2, 3, 4) - it is mentioned that there is an insufficient amount of the data. The overall impression is that the paper is very interesting and has the potential as a useful tool for future readers.
Author Response
Dear Reviewer,
Thank you for reviewing our manuscript. Please find attached the corrected manuscript no. Foods-2544509, entitled: “Sous-vide as an alternative method of cooking to improve the quality of meat: a review”. It has been corrected in line with the comments of reviews.
The text has been checked by proofreader. Revisions were made according to the reviewers’ comments. Changes in the manuscript are indicated in Track revisions.
The answers and comments are attached below. All changes in the manuscript were marked using track changes option in Word and additionally highlighted in color (Blue to Reviewer 1, Green to Reviewer 2, Red to Reviewer 3). We hope that the improved manuscript will find your acceptance for publication. Thank you for your patience and help.
Authors
Comments and Suggestions for Authors
The Article entitled „Sous-vide as an alternative method of cooking to improve the 2 quality of meat: a review” is a review paper considering mechanisms of changes in meat and quality parameters in sous-vide products. It is presented in detail, but there is a need to be checked thoroughly by the authors and removes some sentences - repetition, e.g. introduction, row 27-33, 34-37 etc. Literature data are well and critically presented in this review. There may be a more precise conclusion defined and under the aim of the review. It is not a systematic review and meta-analysis paper, but still, authors should point out the method of collecting/searching string, data presented in tables (1, 2, 3, 4) - it is mentioned that there is an insufficient amount of the data. The overall impression is that the paper is very interesting and has the potential as a useful tool for future readers.
We agree with the Reviewer. We removed these sentences that repetition in Introduction chapter, in row 27-22, and 34-37.
Introduction
We added new sentences according to Reviewer suggestion.
All data presented in this review were summarized from the references, including scientific journals and book chapters. These references were systematically searched against databases: PubMed, Web of Science, Scopus and Google Scholar with a keywords: sous-vide, pork, beef, poultry, cooking yield; sensory quality; nutritional value; microbiological quality. To search for maximum relative references, the keyword was set as “sous-vide and technological process”, and restricted to 2000–2020 years.
Conclusion
We added new sentences according to Reviewer suggestion.
In all the analysed studies, the authors state that the yield of the process depends on the type and properties of the meat and decreases with increasing temperature, while the cooking time is not significant. Similarly, vacuum conditions play a secondary role but affect the shelf life of SV cooked meat. Beef requires a longer cooking process compared to other types of meat, resulting in lower yield. Many studies have shown that cooking SV at 50 °C for extended periods resulted in the lowest cooking losses. And these losses, resulting mainly from changes in the structure and properties of meat proteins, are the highest at temperatures above 60°C. Higher process yield results in higher juiciness.

Reviewer 2 Report
Line 12-13: “These factors act additively, thanks to which they limit unfavorable changes more effectively than individually” this phrase is vague.
I think it is better to add main findings of literature review in the abstract.
Line 27-34: There is a miss and repetition in this paragraph
Line 34-37: Again, there is a miss and repetition in this paragraph, some of phrases are the same as in in the abstract
Line 107: “reduces culinary losses due to lower process parameters, which process parameters the authors mean?
Line 116: Why the authors introduced the word beef followed by full stop. It can be subheading
Line 125: The phrase should be corrected to:” The same results were achieved other authors”
Line 127: “while decreasing re-heating loss” this part of phrase is not clear what decreasing and what is reheating loss?
Line 116: Why the authors introduced the word beef followed by full stop.
Line 139: Why the authors introduced the word beef followed by full stop. It can be subheading
Line 139-143: This paragraph should be restructured
Line 144: the unit of temperature should be corrected and check also the other sites
Line 149: How cook loss increases with post-mortem aging in sous-vide chops
Line 164: 15%, how this value was obtained
Line 165:4%, how this value was obtained
Line 173: in studies, it is a study
Line 219-220: There is only one study about the effect of sous vide on color?
Line 287-289: What do the author mean with this phrase?
Line 299: Higher compound content of what?
Line 396: …….. the increased water exuded.., this phrase is weak
Line 404: solubility of what?
Line 459-460: Compared to what
line 472-473: what is the meaning of synthetic conclusions?
Line 570-572: The authors should clarify the sentence, should indicate the consumption of …..
Line 633: with the process time oscillating between 10 and 50 °C for, why the authors introduced Celsius and time unit
I recommend improving the quality of English language, many parts of the manuscript were written in simple present and active voice.
Author Response
Dear Reviewer,
Thank you for reviewing our manuscript. Please find attached the corrected manuscript no. Foods-2544509, entitled: “Sous-vide as an alternative method of cooking to improve the quality of meat: a review”. It has been corrected in line with the comments of reviews.
The text has been checked by proofreader. Revisions were made according to the reviewers’ comments. Changes in the manuscript are indicated in Track revisions.
The answers and comments are attached below. All changes in the manuscript were marked using track changes option in Word and additionally highlighted in color (Blue to Reviewer 1, Green to Reviewer 2, Red to Reviewer 3). We hope that the improved manuscript will find your acceptance for publication. Thank you for your patience and help.
Authors
Line 12-13: “These factors act additively, thanks to which they limit unfavorable changes more effectively than individually” this phrase is vague. I think it is better to add main findings of literature review in the abstract.
The abstract has been changed as suggested by the reviewer.
After correction:
Sous-vide (SV) is a method of cooking previously vacuum-packed raw materials under strictly controlled conditions of time and temperature. Over the past few years, scientific articles have explored the physical, biochemical, and microbiological properties of SV cooking. In this review, we provide a critical appraisal of SV as an alternative method of meat cooking, including the types of methods, types of SV meat products, and effects of SV parameters on meat quality and mechanisms of transformation taking place in meat during SV cooking. Based on the available data it can be concluded that most research on the SV method refers to poultry. The yield of the process depends on meat type and characteristics, and decreases with increasing temperature, while time duration does not have an impact. Appropriate temperatures in this method make it possible to control changes in products and affect their sensory quality. Vacuum conditions are given a minor role, but they are important during storage. The limited number of studies on the approximate composition of SV meat products makes it challenging to draw summarizing conclusions on this subject. The SV method allows for a higher microbiological quality of stored meat than conventional methods. The literature suggests that the SV method of preparing beef, pork and poultry has many advantages.
Line 27-34: There is a miss and repetition in this paragraph
Line 34-37: Again, there is a miss and repetition in this paragraph, some of phrases are the same as in in the abstract.
Until 2011, it was mainly used to extend the shelf life of food products with the help of the so-called combined methods, which integrate several parallel fixing factors: vacuum conditions, low temperature, and extended process time.
After correction
Line 107: “reduces culinary losses due to lower process parameters, which process parameters the authors mean?
The sous-vide method reduces culinary losses due to lower process parameters, vacuum packaging, and reduced water loss from meat, shrinkage, and juice loss [38, 42, 60, 61].
After correction
The sous-vide method reduces culinary losses due to lower process temperatures, vacuum packaging, and reduced water loss from meat, shrinkage, and juice loss [38, 42, 60, 61].
Line 116: Why the authors introduced the word beef followed by full stop. It can be subheading
Yes, it is the subtitle.
Line 125: The phrase should be corrected to:” The same results were achieved other authors”
We agree with the Reviewer and corrected it.
Line 127: “while decreasing re-heating loss” this part of phrase is not clear what decreasing and what is reheating loss?
We agree with the Reviewer, and to better understand this sentence we remove „while decreasing re-heating loss”.
Cooking at higher temperatures for longer periods of time increased sous-vide cooking loss while decreasing re-heating loss.
Line 139: Why the authors introduced the word beef followed by full stop. It can be subheading
Yes, it is the subtitle, but it is not needed here.
Line 139-143: This paragraph should be restructured
We agree for the Reviewer. Thank you for this suggestion, we have combined two paragraphs for better understanding of the text. We changed the structure of this paragraph
Beef.The sous-vide process of beef meat (whole muscle or slices 1.5–4.0 cm thick) was carried out by the authors in the temperature range of 50–100 °C and for 30 to 1080 min (for 18 h). The yield of the obtained products was affected by the temperature and time of the process, with the temperature being the decisive factor. The increase in each of these parameters resulted in a decrease in the yield of the process (Table 1). Extending the sous-vide process at a given temperature had a lesser effect on the water content of the product.
Beef requires a longer heat treatment process compared to other types of meat, resulting in lower efficiency. Only during sous-vide processing at 50 °C does it achieve 89.2-91.7% efficiency, similar to chicken processing (Table 1). However, higher temperatures and longer thermal treatments lead to lower yields of 61.1–75.1%. Diverse process parameters result from meat structure, fat content, and portion size (size and thickness).
After correction
Beef.The sous-vide processing of beef meat (whole muscle or slices 1.5–4.0 cm thick) was carried out by the authors in the temperature range of 50–100 °C and for 30 to 1080 min (for 18 h). Diverse process parameters result from meat structure, fat content, and portion size (size and thickness). Beef requires a longer heat treatment process compared to other types of meat, resulting in lower yield. The yield of the obtained products was affected by the temperature and duration of the process, with the temperature being the decisive factor. An increase in either of these parameters resulted in a decrease in the yield of the process (Table 1). Only during sous-vide processing at 50 °C does it achieve 89.2-91.7% yield, similar to chicken processing (Table 1). However, higher temperatures and longer thermal treatments lead to lower yields of 61.1–75.1%. Extending the sous-vide process at a given temperature had a lesser effect on the water content of the product.
Line 144: the unit of temperature should be corrected and check also the other sites
Thank You, we corrected this.
Line 149: How cook loss increases with post-mortem aging in sous-vide chops
Thank You. We have to correct English because this sentence is not understandable.
Cook loss increases with postmortem aging in sous-vide chops [21].
After correction
The cooking loss of chops, when prepared with the sous-vide method, increases with the duration of postmortem aging of meat [21].
Line 164: 15%, how this value was obtained. Line 165:4%, how this value was obtained.
Thank You. This is our mistake. We changed this sentence.
The process efficiency decreased with the increase in the process temperature by about 15%, while the differences in the water content of the raw materials subjected to different methods differed by about 4%.
After correction
As the process temperature increased up to 100 °C and more, the yield of meat increased from 79.4 to 65.8% for beef, from 83.3 to 58.7% for pork, and from 93.4 to 71% for poultry (about 13.6–24.6% average). However, differences in the water content of the raw materials subjected to various methods differed in beef from 70 to 53%, in pork from 68.4 to 58.4%, and in poultry from 74.9 to 53.9% (about 10–17% average).
Line 173: in studies, it is a study
Line 219-220: There is only one study about the effect of sous vide on color?
Line 287-289: What do the author mean with this phrase?’
Thank you for this comments, we corrected this.
After correction
The multi-stage sous-vide method of meat preparation, which involves of thermal treatment at increasing temperatures in stages, has been increasingly emphasized [65].
Line 299: Higher compound content of what?
Thank you for this comments, we corrected this.
Line 396: …….. the increased water exuded.., this phrase is weak
We agree with the Reviewer and corrected this sentence.
Meat cooked at 60 °C has higher adhesiveness than meat cooked at 80 °C due to the increased water exuded, but vacuum conditions don't affect the texture.
After correction
Meat cooked at 60 °C has higher adhesiveness than meat cooked at 80 °C. The reason for this is the increased release of water at 80C, which, however, does not change the texture.
Line 404: solubility of what?
Line 459-460: Compared to what
Line 472-473: what is the meaning of synthetic conclusions?
Line 570-572: The authors should clarify the sentence, should indicate the consumption of
Line 633: with the process time oscillating between 10 and 50 °C for, why the authors introduced Celsius and time unit
We agree with the Reviewer and corrected this sentence.
Comments on the Quality of English Language
I recommend improving the quality of English language, many parts of the manuscript were written in simple present and active voice.
Text was corrected by proofreader.

Reviewer 3 Report
The manuscript reviewed the sous-vide cooking method to improve the quality of meat in recent years. There are some parts that need to improve. The specific comments are as follows:
Line 52: "distinguish" should be "distinguished"
Line 59: "Multi-Stage Sous-Vide Cooking" should be "multi-stage sous vide cooking"
It will be interesting if it adds the principle of sous-vide cooking as a new section. Especially the "H" and "P" in Table 1, which is not clear what they are used for.
Line 89: it stated the experiments carried out by various authors, please provide the references. The same problem also occurred in Line 190
Please use the three-line table
Table 4: I am wondering why the microbial count was minus for the first two rows, and why the account become less after several days of storage than the fresh day, usually, it increases after several days of storage, please explain the reason.
There are no figures, and it will be nice if some content can be summarized using figures.
There are some places that need to revise the English
Author Response
Dear Reviewer,
Thank you for reviewing our manuscript. Please find attached the corrected manuscript no. Foods-2544509, entitled: “Sous-vide as an alternative method of cooking to improve the quality of meat: a review”. It has been corrected in line with the comments of reviews.
The text has been checked by proofreader. Revisions were made according to the reviewers’ comments. Changes in the manuscript are indicated in Track revisions.
The answers and comments are attached below. All changes in the manuscript were marked using track changes option in Word and additionally highlighted in color (Blue to Reviewer 1, Green to Reviewer 2, Red to Reviewer 3). We hope that the improved manuscript will find your acceptance for publication. Thank you for your patience and help.
Authors
The manuscript reviewed the sous-vide cooking method to improve the quality of meat in recent years. There are some parts that need to improve. The specific comments are as follows:
Line 52: "distinguish" should be "distinguished"
Thank You. We corrected this word.
Line 59: "Multi-Stage Sous-Vide Cooking" should be "multi-stage sous vide cooking"
It will be interesting if it adds the principle of sous-vide cooking as a new section.
Probably, it would be interesting if we added the principle of sous-vide cooking as a new section, but that applies almost all of this text. We decided not to change it.
Especially the "H" and "P" in Table 1, which is not clear what they are used for.
We explained this in lines 102-105.
Line 89: it stated the experiments carried out by various authors, please provide the references. The same problem also occurred in Line 190
Table 4: I am wondering why the microbial count was minus for the first two rows, and why the account become less after several days of storage than the fresh day, usually, it increases after several days of storage, please explain the reason.
In the table this is not minus, but dash.
The account become less after several days as explained authors:
“This result might be due to the inhibitory effect of vacuum packaging on microorganism growth under anaerobic conditions. Similar results were observed in the studies by Wang et al. (2004) and Ramane et al. (2010), which reported that vacuum packaging prevented spoilage of the product because of the hurdle effect of anaerobic conditions and some chemical reactions. The coliform count in the S group was significantly (p<0.05) lower than that in the R group”.
There are no figures, and it will be nice if some content can be summarized using figures.
In this article, we wanted to explain the effect of the sous-vide method on meat quality. The article is long and we would prefer not to add additional Figures.
Comments on the Quality of English Language
There are some places that need to revise the English
Text was corrected by proofreader.

Round 2
Reviewer 2 Report
No comments